# EA.hy926 Cells and HUVECs Share Similar Senescence Phenotypes but Respond Differently to the Senolytic Drug ABT-263

**DOI:** 10.3390/cells11131992

**Published:** 2022-06-21

**Authors:** Ibrahim Y. Abdelgawad, Kevin Agostinucci, Somia G. Ismail, Marianne K. O. Grant, Beshay N. Zordoky

**Affiliations:** Department of Experimental and Clinical Pharmacology, College of Pharmacy, University of Minnesota, Minneapolis, MN 55455, USA; abdel217@umn.edu (I.Y.A.); agost031@umn.edu (K.A.); sismail@umn.edu (S.G.I.); grant032@umn.edu (M.K.O.G.)

**Keywords:** endothelial cells, doxorubicin, senescence, senolytics, ABT-263, BCL-2 family

## Abstract

Doxorubicin (DOX) induces endothelial cell (EC) senescence, which contributes to endothelial dysfunction and cardiovascular complications. Senolytic drugs selectively eliminate senescent cells to ameliorate senescence-mediated pathologies. Previous studies have demonstrated differences between immortalized and primary EC models in some characteristics. However, the response of DOX-induced senescent ECs to senolytics has not been determined across these two models. In the present work, we first established a comparative characterization of DOX-induced senescence phenotypes in immortalized EA.hy926 endothelial-derived cells and primary human umbilical vein EC (HUVECs). Thereafter, we evaluated the senolytic activity of four senolytics across both ECs. Following the DOX treatment, both EA.hy926 and HUVECs shared similar senescence phenotypes characterized by upregulated senescence markers, increased SA-β-gal activity, cell cycle arrest, and elevated expression of the senescence-associated secretory phenotype (SASP). The potentially senolytic drugs dasatinib, quercetin, and fisetin demonstrated a lack of selectivity against DOX-induced senescent EA.hy926 cells and HUVECs. However, ABT-263 (Navitoclax) selectively induced the apoptosis of DOX-induced senescent HUVECs but not EA.hy926 cells. Mechanistically, DOX-treated EA.hy926 cells and HUVECs demonstrated differential expression levels of the BCL-2 family proteins. In conclusion, both EA.hy926 cells and HUVECs demonstrate similar DOX-induced senescence phenotypes but they respond differently to ABT-263, presumably due to the different expression levels of BCL-2 family proteins.

## 1. Introduction

The population of cancer survivors has remarkably increased in recent years due to significant advances in the diagnosis and treatment of cancer patients [1]. With the increase in survival rate, the long-term adverse effects of cancer treatments become more evident. Accelerated aging and premature frailty are among the long-term adverse complications induced by chemotherapy [2,3]. Additionally, many cancer treatments can adversely affect the cardiovascular system and increase the susceptibility of cancer survivors to develop premature cardiovascular complications [4]. Although several molecular mechanisms have been proposed to explain cancer-therapy-induced cardiovascular complications, the role of therapy-induced premature aging in mediating these cardiovascular complications has recently been recognized [2]. Cellular senescence, one pillar of aging, is a state of stable cell cycle arrest in which cells demonstrate morphological and functional alterations [5]. The accumulation of senescent cells contributes to the development of age-related diseases [6]. Multiple cancer treatments induce senescence in cardiovascular cells via different mechanisms, suggesting a role of cellular senescence in cancer-therapy-induced cardiovascular complications [2]. Doxorubicin (DOX) is one of the most widely used chemotherapeutic agents in both pediatric and adult clinical oncology. Despite its effectiveness in a wide range of cancers, DOX’s clinical utility is limited due to its cardiotoxicity. In addition to cardiotoxicity, DOX also induces vascular toxicity, characterized by endothelial dysfunction and vascular aging [7,8,9]. Interestingly, low doses of DOX induce endothelial senescence and premature vascular aging rather than apoptosis [10,11,12,13].

The elimination of senescent cells can be a promising strategy to mitigate cancer-therapy-induced endothelial senescence [14]. The last decade has witnessed the development of senolytics, which are a novel class of drugs that selectively induce the apoptosis of senescent cells by interfering with the pro-survival pathways expressed by senescent cells. Several senolytics including quercetin, fisetin, and ABT-263 (navitoclax) have been shown to exert senolytic effects in human umbilical vein endothelial cells (HUVECs) [15,16,17]. HUVEC is a primary cell line that is a reference in vitro model for studying the senescence of endothelial cells (ECs). However, no previous studies have reported the senolytic effects of these compounds in immortalized EC cell lines such as EA.hy926 cell line, which is a hybrid cell line that results from the fusion of HUVECs and A549 lung carcinoma cells [18]. Primary HUVECs are characterized by their limited life span, difficulty to maintain, and batch-to-batch variation, which can affect the reproducibility of the results [19]. On the other hand, immortalized EA.hy926 cells grow faster, are easier to handle, and yield reproducible results [18]. However, due to their hybrid nature, EA.hy926 cells may have some different characteristics compared to primary ECs. Indeed, a recent study demonstrated that EA.hy926 cells are deficient in some endothelial properties and have a transcriptomic profile that differs from primary ECs [20]. In the current study, we first established a comparative characterization of DOX-induced senescence phenotypes across EA.hy926 cells and HUVECs and then compared the senolytic effects of multiple senolytics in both cell lines. We demonstrate that EA.hy926 cells and HUVECs show similar molecular and phenotypic characteristics of DOX-induced senescence. Only ABT-263 exerted selective senolytic activity in DOX-induced senescent HUVECs, but not in EA.hy926 cells.

## 2. Materials and Methods

### 2.1. Materials

Dulbecco’s modified Eagle’s medium (DMEM), fetal bovine serum, phosphatase inhibitor, 3-(4,5-dimethylthiazol-2-yl)-2,5-diphenyl-2H-tetrazolium bromide (MTT), doxorubicin (DOX), dasatinib, and quercetin were purchased from MilliporeSigma (St. Louis, MO, USA). Vascular cell basal medium and endothelial cell growth kit–VEGF were purchased from the American Type Culture Collection (ATCC) (Manassas, FL, USA). A Pierce™ bicinchoninic acid (BCA) protein assay kit, Pierce™ ECL substrate, High-Capacity cDNA Reverse Transcription Kit, penicillin–streptomycin, SYBR™ Green Master Mix, RNase A, and propidium iodide were purchased from ThermoFisher (Waltham, MA, USA). ABT-263 (Navitoclax), ABT-199 (Venetoclax), and fisetin were purchased from MedChemExpress (Monmouth Junction, NJ, USA); a protease inhibitor cocktail was purchased from Sigma (St. Louis, MO, USA); PCR primers were obtained from IDT (Integrated DNA Technologies) (Coralville, IA, USA); a senescence β-galactosidase (SA-β-gal) staining kit was purchased from Cell Signaling (Danvers, MA, USA); and nitrocellulose membranes were purchased from BioRad (Hercules, CA, USA).

Primary mouse antibodies against p53 (catalog #2524) and primary rabbit antibodies against p21 (catalog #2947), cyclin D1 (catalog #2978), cleaved caspase-3 (catalog #9664), caspase-3 (catalog #9662), cleaved PARP (catalog #5625), PARP (catalog #9542), BCL-2 (catalog #4223), BCL-xL (catalog #2764), BCL-W (catalog #2724), BAK (catalog #12105), BAX (catalog #2774), and alpha-tubulin (catalog #2144) were purchased from Cell Signaling. HRP-conjugated horse anti-mouse secondary antibodies (catalog #7076) were purchased from Cell Signaling and HRP-conjugated goat anti-rabbit secondary antibodies (catalog #111-035-144) were purchased from Jackson ImmunoResearch (West Grove, PA, USA).

### 2.2. Cell Culture

The EA.hy926 human-endothelial-derived cell line and human umbilical vein endothelial cells (HUVECs) were purchased from ATCC. EA.hy926 cells were cultured in DMEM supplemented with fetal bovine serum (10% *v*/*v*), penicillin (100 U/mL), and streptomycin (100 μg/mL). HUVECs were cultured in vascular cell basal medium supplemented with endothelial cell growth kit–VEGF, including rh VEGF (5 ng/mL), rh EGF (5 ng/mL), rh FGF basic (5 ng/mL), rh IGF(15 ng/mL), L-glutamine (10 mM), heparin sulfate (0.75 U/mL), hydrocortisone (1 µg/mL), ascorbic acid (50 µg/mL), fetal bovine serum (2%), penicillin (10 U/mL), and streptomycin (10 µg/mL). Both cell lines were maintained in 75 cm^2^ tissue-culture-treated flasks in a 5% CO_2_ humidified incubator at 37 °C. Media were changed every other day and cells were subcultured when they became 80% confluent.

### 2.3. Cell Treatments

All stock solutions were prepared before starting cell treatments. DOX stock solutions were prepared by dissolving DOX in the DMEM or vascular cell basal media. To characterize the senescence phenotypes, EA.hy926 cells and HUVECs were seeded into 6-well plates and incubated with increasing concentrations of DOX (0.1 μM, 0.2 μM, and 0.5 μM) for 24 h. Thereafter, the DOX was removed, cells were washed with phosphate-buffered saline (PBS), then incubated in DOX-free growth media for another 72 h. The 0.5 μM concentration of DOX was chosen for subsequent experiments because it was associated with the highest induction of senescence markers.

For the assessment of senolytic activity experiments, senescence was first induced by treating cells with 0.5 µM DOX for 24 h followed by 72 h incubation in DOX-free growth media. Then, cells were treated with increasing concentrations of each senolytic for an additional 24 h followed by measuring the cell viability. To assess the role of apoptotic cell death in mediating the observed senolytic activity of ABT-263, senescence was first induced by treating cells with 0.5 µM DOX for 24 h followed by 72 h incubation in DOX-free growth media. Then, cells were treated with 0.1 µM ABT-263 for 6 h followed by the assessment of apoptotic markers. Stock solutions of all senolytics were prepared by dissolving in 100% dimethylsulfoxide (DMSO). Control wells were treated with an equal volume of DMSO, which did not exceed a volume of 1% of the medium.

### 2.4. Protein Extraction and Western Blotting

EA.hy926 cells and HUVECs were grown in 6-well plates and subjected to the treatments described above. Cells were washed with PBS and harvested in a lysis buffer containing 20 mM Tris, 100 mM sodium fluoride, 10 mM sodium pyrophosphate, 5 mM EDTA, and 1% NP-40 in the presence of protease and phosphatase inhibitors. Cells were lysed by passing through a 28 gauge needle 10 times and the lysate was centrifuged at 2000× *g* for 10 min at 4 °C. Thereafter, the supernatant was collected for Western blotting. The protein concentration was measured using the Pierce™ bicinchoninic acid (BCA) protein assay kit according to the manufacturer’s instructions. Cell homogenates were denatured by boiling at 100 °C for 5 min in sodium dodecyl sulfate (SDS)–polyacrylamide gel electrophoresis (PAGE) loading buffer. Thereafter, 20–30 μg homogenates were separated on 12% SDS-PAGE gels and transferred to nitrocellulose membranes. The blots were then blocked at room temperature for 1 h with 5% skim milk powder in Tris-buffered saline (20 mM Tris, 150 mM NaCl, pH 7.4) with 0.05% (*v*/*v*) Tween-20 (TBST). Following blocking, blots were incubated overnight at 4 °C with primary antibodies diluted 1:1000 in 1% milk solution in TBST. Blots were then washed in TBST and incubated for 1 h at room temperature with horseradish peroxidase (HRP)-conjugated secondary antibodies diluted in blocking buffer, then washed with TBST. Blots were visualized using Pierce™ ECL substrate according to the manufacturer’s instructions. An Amersham Imager 600 (Cytiva, Marlborough, MA, USA) was used to visualize the chemiluminescence. For blot stripping, blots were incubated for 30 min in stripping buffer (62.5 mM Tris, 2% SDS, 18mM 2-mercaptoethanol, pH 6.7) at 50 °C, washed, blocked, and probed as previously described. ImageJ software (National Institutes of Health, Bethesda, MD, USA) was used to quantify band intensities using alpha-tubulin protein levels as normalizing loading controls. Phospho-protein band intensities were measured relative to the respective total protein level. In some experiments, the blots were cut at separate molecular weight marks, thereby allowing the same blot to be incubated with more than one primary antibody, reducing the need for blot stripping. Uncropped Western blot images are shown in Appendix A.

### 2.5. Senescence-Associated β-galactosidase (SA-β-gal) Assay

The SA-β-gal staining kit was used to stain senescent cells according to the manufacturer’s protocol. To avoid false-positive staining in non-treated cells, the staining incubation time and pH were optimized for EA.hy926 cells and HUVECs in pilot experiments. We found that the optimal conditions differed in each cell line. EA.hy926 cells were incubated with the stain for 16 h at pH 6.5, and HUVECs were stained for 6 h at pH 6. The percentage of SA-β-gal-positive cells was then calculated by counting the number of stained cells relative to total cell number in random fields (at least 100 cells) following capture using a bright-field microscope (4× objective lens).

### 2.6. Cell Cycle Analysis

A cell cycle analysis was performed using propidium iodide (PI) staining. Following the specified treatments, cells were harvested using trypsin and washed twice with cold PBS. Cells were suspended in PBS at a concentration of 1–2 × 10^6^ cells/mL and fixed in 70% ethanol for 45 min on ice followed by incubation overnight at −20 °C. Cells were centrifuged at 300× *g* for 10 min at 4 °C and the pellet was suspended in 1 mL PI master mix (40 µL PI, 10 µL RNase A, and 950 µL PBS). Tubes were wrapped in aluminum foil and incubated at 37 °C for 30 min. The cell cycle was analyzed by measuring the DNA content using the FACSCanto system (BD Biosciences, Franklin Lakes, NJ, USA). At least 30,000 events were collected in each analysis at a low event rate. Data were further analyzed using FACSdiva software (BD Biosciences).

### 2.7. RNA Extraction and Real-Time PCR

Following the treatments described above for the specified periods, the total cellular RNA was isolated using TRIzol^®^ reagent (Life Technologies, Carlsbad, CA, USA) according to the manufacturer’s instructions. The total RNA was quantified by using a Nanodrop 8000 spectrophotometer (Thermo Fisher Scientific, Waltham, MA, USA). Thereafter, the first-strand cDNA was synthesized from 1.5 μg of total RNA using the high-capacity cDNA reverse transcription kit according to the manufacturer’s instructions. To measure specific mRNA expression levels, a real-time polymerase chain reaction (PCR) was carried out in 384-well optical plates in a final volume of 20 μL reaction mix (1 μL cDNA sample, 0.025 μL 30 μM forward primer, 0.025 μL 30 μM reverse primer, 10 μL SYBR Green Mastermix, and 8.95 μL of nuclease-free water) using a QuantStudio 5 instrument (Applied Biosystems, Foster City, CA, USA). The following thermocycling amplification conditions were used: 95 °C for 10 min, 40 PCR cycles of denaturation at 95 °C for 15 s, then annealing–extension at 60 °C for 1 min. At the end of each cycle, a melting curve analysis was implemented to ensure the specificity of the primers used. The sequences of the primer pairs are listed in Table 1. The primers selected in the current study were checked with the Primer-BLAST online tool. The ΔΔCt method was used to identify DOX-induced gene expression changes. Data were normalized to endogenous beta-2 microglobulin (B2M) and expressed relative to control cells.

### 2.8. Assessment of Senescence-Associated Secretory Phenotype (SASP) Factors in Cell Culture Media

Cell culture media were collected from both EA.hy926 cells and HUVECs following the specified treatments. Thereafter, media were stored at −80 °C until use. Supernatants were analyzed by the Cytokine Reference Laboratory (University of Minnesota, Minneapolis, MN, USA) for human-specific interleukin 6 (IL-6), interleukin 8 (IL-8), tumor necrosis factor alpha (TNF-α), interleukin 1 beta (IL-1β), and monocyte chemoattractant protein-1 (MCP-1) using the Luminex platform and conducted as a multiplex. Cytokines were analyzed according to the manufacturer’s guidelines by lab personnel, who were blinded to the experimental design. Briefly, fluorescent color-coded beads coated with a particular capture antibody were used to treat each sample. Following incubation and washing, a biotinylated detection antibody was added followed by phycoerythrin-conjugated streptavidin. A Luminex instrument (Bioplex 200, Bio-Rad Laboratories, Inc., Hercules, CA, USA) was used to read the beads. Samples were run in duplicate and values were interpolated from 5-parameter equipped standard curves. Cytokine values were normalized to the protein content (determined by BCA) and reported as fold changes vs. control cells.

### 2.9. Cell Viability Assay

The cell viability was measured by MTT assay, which measures the capacity of viable cells to convert MTT to formazan crystals using reducing enzymes. EA.hy926 cells and HUVECs were seeded in 96-well cell culture plates and treated as described above in cell treatments for senolytic experiments. Following treatments, the media were removed, then 90 μL of media containing MTT (5 mg/mL) was added and the plate was incubated at 37 °C for 2 h. Thereafter, the MTT solution was removed and 200 μL of isopropyl alcohol was added to each well, followed by shaking for 1 h at room temperature. The color intensity in each well was measured at a wavelength of 550 nm using a Biotek 800TS microplate reader (Agilent, Santa Clara, CA, USA). The cell viability of vehicle- and DOX-treated groups was calculated as a percentage relative to the control wells of each treatment group, which is considered as 100% viable cells.

### 2.10. Statistical Analysis

A data analysis was performed using GraphPad Prism software (version 8.3.0, La Jolla, CA, USA), and the data are presented as the mean ± standard error of the mean (SEM). Comparisons between two groups were analyzed using an unpaired t-test. For comparisons of three or more groups, a one-way analysis of variance (ANOVA) was performed followed by Dunnet’s multiple comparison test. The cell cycle analysis was analyzed via two-way ANOVA followed by Sidak’s post hoc test. Here, each *p* value < 0.05 was considered statistically significant.

## 3. Results

### 3.1. Doxorubicin Induced the Expression of Senescence Markers in EA.hy926 Cells and HUVECs

To assess the senescence phenotype in EA.hy926 cells and HUVECs, both cell lines were treated with increasing concentrations (0.1–0.5 µM) of doxorubicin (DOX). Following 24 h treatment with DOX, cells were washed and incubated in DOX-free media for an additional 72 h. Thereafter, the protein expression of the senescence markers p53, p21, and cyclin D1 was assessed. The p53/p21 pathway is activated by the DNA damage response and is involved in cell cycle regulation [23]. Cyclin D1 is another cell cycle regulator and a downstream target of p53 [24]. In EA.hy926 cells, DOX treatment resulted in a concentration-dependent upregulation of the expression of p53 (Figure 1A), p21 (Figure 1B), and cyclin D1 (Figure 1C). The highest concentration of DOX (0.5 µM) caused a remarkable 17-fold increase in p53 expression and significant 2.4-fold increases in p21 and cyclin D1. A similar concentration-dependent upregulation of senescence markers was elicited by DOX in HUVECs. The highest concentration of DOX (0.5 µM) caused a significant 6.9-fold increase in p53 (Figure 1D), a 3.7-fold increase in p21 (Figure 1E), and a 1.7-fold increase in cyclin D1 (Figure 1F).

### 3.2. Both EA.hy926 Cells and HUVECs Demonstrated Increased SA-β-gal Activity and Cell Cycle Arrest

The senescence phenotype is characterized by cellular alterations, including enhanced senescence-associated beta-galactosidase (SA-β-gal) activity and cell cycle arrest. We evaluated these changes in EA.hy926 and HUVEC cell lines following treatment with 0.5 µM DOX as described in the methods section. As illustrated in Figure 2A, the percentage of SA-β-gal-positive cells was markedly increased in EA.hy926 cells treated with DOX compared with control cells (75% vs. 9%, respectively). In HUVECs (Figure 2B), a more modest increase in SA-β-gal activity was observed following DOX treatment compared to control cells (47.3% vs. 7.3%, respectively). The cell cycle analysis following the treatment of EA.hy926 cells with 0.5 µM DOX demonstrated a remarkable increase in the G2/M arrested population from 4.1% in the control cells to 60.9% in the DOX-treated cells (Figure 2C). A less remarkable, but still significant, increase in the percentage of cells in the G2/M phase was demonstrated in HUVECs, where the G2/M population increased from 31.6% in control cells to 41.9% in DOX-treated cells (Figure 2D).

### 3.3. Doxorubicin Induced the Expression of Senescence-Associated Secretory Phenotype (SASP) Factors in EA.hy926 Cells and HUVECs in a Similar Manner

The expression of SASP factors is another important feature of cellular senescence. The expression levels of SASP factors have previously been shown to differ between different cell types [25,26]. Therefore, we measured changes in the gene expression levels of several SASP factors, including IL-6, C-X-C chemokine ligand 1 (CXCL1), and C-X-C chemokine ligand 8 (CXCL8), induced by 0.5 µM DOX treatment in both EC lines. In EA.hy926 cells, the DOX treatment was associated with significant 2-, 3-, and 1.8-fold increases in IL-6, CXCL1, and CXCL8, respectively (Figure 3A). Similarly, the DOX treatment in HUVECs resulted in a significant 6.4-fold increase in IL-6 and a 3.5-fold increase in CXCL1 (Figure 3B). However, the CXCL8 expression was not significantly changed in HUVECs (Figure 3B). Since SASP includes soluble factors that can be secreted and can affect neighboring cells, we measured the concentrations of IL-6, TNF-α, IL-8, IL-1ß, and MCP-1 in the culture media of both cell lines following DOX treatment. In agreement with the observed increase in the gene expression of SASP factors, the DOX treatment also resulted in the increased secretion of SASP factors in the culture media. In EA.hy926 cells, significant 2.5-, 1.8-, and 1.7-fold increases in IL-6, IL-8, and TNF-α, respectively, were identified in the culture media (Figure 4A). Moreover, an increasing but not significant trend was observed in IL-1ß and MCP-1 (Figure 4A). DOX-treated HUVECs demonstrated a similar upregulation of SASP factors in the culture media. This included significant 3-, 2.1-, 1.5-, and 1.5-fold increases in IL-6, TNF-α, IL-1ß, and MCP-1, respectively (Figure 4B). The IL-8 expression was increased by 2-fold, although this was not found to be significant (Figure 4B).

### 3.4. ABT-263 Demonstrated Differential Senolytic Activity in EA.hy926 Cells and HUVECs

An increasing number of drugs have been demonstrated to exhibit senolytic activity. Importantly, previous studies have demonstrated that the activity of senolytics depends on the cell type [16,17,27]. For example, quercetin, fisetin, and ABT-263 have been shown to exert senolytic effects in senescent HUVECs but not in preadipocytes. However, dasatinib was more effective in eliminating senescent preadipocytes than HUVECs [15,16,17]. Therefore, we sought to screen the senolytic activity of these senolytics in both EA.hy926 cells and HUVECs by evaluating their effects on cell viability. In agreement with the previous study, dasatinib failed to induce senolytic activity and decreased the viability of untreated and DOX-treated senescent cells to similar degrees in both EA.hy926 cells and HUVECs (Figure 5A). Quercetin has demonstrated a very modest senolytic effect in EA.hy926 cells but not in HUVECs (Figure 5B). Fisetin failed to exert senolytic activity in both EA.hy926 cells and HUVECs (Figure 5C). Only ABT-263 exhibited a remarkably differential activity in these cell lines (Figure 5D). In EA.hy926 cells, ABT-263 did not selectively target senescent cells and decreased the viability of both DOX-induced senescent cells and non-senescent control cells in a similar manner (Figure 5D). In contrast, ABT-263 showed high senolytic activity against DOX-induced senescent HUVECs and remarkably decreased the viability of DOX-induced senescent cells compared to non-senescent cells (Figure 5D).

Since apoptosis is the main driver of the senolytic activity of ABT-263, we assessed the effects of ABT-263 on the cleavage of two apoptotic markers in EA.hy926 cells and HUVECs. In EA.hy926 cells, ABT-263 significantly induced the cleavage of caspase-3 in both control- and DOX-treated cells (Figure 6A). Cleaved-PARP was also increased by ABT-263 in control- and DOX-treated cells (Figure 6B), although this was not statistically significant. In HUVECs, ABT-263 selectively triggered the apoptosis of DOX-induced senescent cells, as demonstrated by the significant upregulation of cleaved-caspase 3 (Figure 6C) and cleaved-PARP (Figure 6D) only in the DOX-treated cells, which was in agreement with the cell viability results.

### 3.5. EA.hy926 Cells and HUVECs Demonstrated Differential Expression of the BCL-2 Family following DOX Treatment

ABT-263 is a non-selective inhibitor of the BCL-2 family (BCL-2, BCL-xL, and BCL-W) [28]. To elucidate the underlying mechanisms of the differential selectivity of ABT-263, we analyzed the protein expression levels of the BCL-2 family across both cell lines. In EA.hy926 cells, the anti-apoptotic proteins BCL-xL and BCL-W were not upregulated following DOX-treatment (Figure 7A). Additionally, the expression of the anti-apoptotic BCL-2 protein was undetectable in EA.hy926 cells (Figure 7A). In contrast, DOX significantly upregulated the protein expression levels of the anti-apoptotic members BCL-2, BCL-xL, and BCL-W in HUVECs (Figure 7B). The same trend was also observed for apoptotic members of the BCL-2 family, namely BAX and BAK. There were no changes in their expression following DOX treatment in EA.hy926 cells, while they were significantly upregulated in DOX-treated HUVECs (Figure 7A,B).

To further confirm our findings that EA.hy926 cells do not express BCL-2 protein, we treated both EA.hy926 cells and HUVECs with venetoclax, a BCL-2-selective inhibitor with lower affinity to BCL-xL and BCL-W. We found that in EA.hy296 cells, venetoclax had a modest effect on the viability of both untreated and DOX-treated senescent cells, with a slightly more lethal effect on DOX-induced senescent cells (65% vs. 55% viability, respectively) (Figure 8). However, venetoclax remarkably reduced the viability of both untreated and DOX-treated cells (<5% viability in both non-senescent and DOX-induced senescent HUVECs), confirming that BCL-2 protein is differentially expressed in these two cell lines (Figure 8).

## 4. Discussion

Cardiovascular senescence induced by doxorubicin (DOX) has recently been proposed as an important mechanism that can mediate the delayed cardiovascular complications in cancer survivors [29]. A plethora of in vitro studies demonstrate that DOX induces senescence in different types of cardiovascular cells, including endothelial cells (ECs), ventricular myocytes, vascular smooth muscle cells, endothelial progenitor cells, and cardiac progenitor cells [2]. Interestingly, a previous in vivo study demonstrated that ECs comprised the majority of the cardiac senescent cell population in DOX-treated mice and that the genetic elimination of senescent cells abrogated DOX-induced cardiac dysfunction [29]. This strongly suggests that senescent ECs are key players in delayed DOX-induced cardiac dysfunction. Importantly, senescent ECs exhibit functional alterations that lead to endothelial dysfunction, including a decrease in vasodilation response and diminished angiogenic activity [14]. Additionally, senescent ECs demonstrated higher SASP expression compared to other cell types [30]. These lines of evidence suggest that endothelial senescence can contribute to cardiovascular complications in DOX-treated patients. Thus, the characterization of EC senescence is the first step to identify therapeutic approaches that may attenuate DOX-induced senescence and prevent premature cardiovascular complications. Although several studies have described DOX-induced senescence in HUVECs, DOX-induced senescence in EA.hy926 cells has not been fully characterized. In this study, we performed an extensive comparative characterization of DOX-induced senescence phenotypes in both HUVECs and EA.hy926 cell lines, including the expression of senescence markers, SA-β-gal activity, cell cycle arrest, the expression of SASP factors, and their response to the senolytics.

We found that DOX induced the senescence markers p53 and p21 in HUVECs, in agreement with previous studies [31,32]. This was also consistent with another study of DOX-induced senescence in human aortic endothelial cells, another type of primary ECs [12]. Interestingly, EA.hy926 cells showed a similar pattern of induction of both p53 and p21, in agreement with a study by Ghosh et al. [33]. We also demonstrated an increase in another senescence marker, cyclin D1, in both cell lines. DOX has been shown to upregulate cyclin D1 in several cell types [34,35], although, to our knowledge, this is the first time it has been reported in ECs. Cellular senescence is also characterized by the upregulation of SA-β-gal activity and cell cycle arrest. Both EA.hy926 cells and HUVECs displayed increased SA-β-gal activity following DOX treatment. Notably, EA.hy926 cells demonstrated higher SA-β-gal activity than HUVECs following DOX treatment. DOX has been demonstrated to induce G2/M arrest in different types of cardiovascular cells, including endothelial progenitor cells [36] and vascular smooth muscle cells [37]. Consistent with these observations, our results showed G2/M arrest in DOX-treated HUVECs and EA.hy926 cells.

The senescence-associated secretory phenotype (SASP) is another important characteristic of senescent cells that encompasses multiple factors, including inflammatory cytokines, chemokines, growth factors, and extracellular matrix proteins [38]. Importantly, the accumulation of SASP contributes to the pathophysiological effects of senescence and activates a low-grade inflammatory state, called “inflammaging”, contributing to multiple age-related diseases. In the current study, DOX upregulated multiple SASP factors in ECs, including IL-6, IL-8, CXCL1, and TNF-α. IL-6 and IL-8 are among the most studied pro-inflammatory SASP factors that increase in senescent ECs [39] and contribute to cardiovascular complications [14]. The increased expression of SASP factors activates the immune response, which leads to the invasion of monocytes into the vessel wall and initiates plaque formation [40]. IL-6 has been shown to increase the secretion of adhesion molecules in ECs and exacerbate atherosclerosis in mice [41]. Additionally, MCP-1 and IL-8 facilitate the attachment of monocytes on the surfaces of ECs during atherosclerosis [42]. Beyond the cardiovascular system, ECs are an essential component in the tumor microenvironment. Therefore, SASP components secreted by senescent ECs can also affect the tumor microenvironment and increase a cancer’s progression and aggressiveness [31]. Indeed, a recent study demonstrates that adding culture media from senescent HUVECs to cancer cells increased the cell proliferation, migration, and invasion, mainly through CXCL11 [43].

Senolytics are a recently developed class of drugs that can specifically eliminate senescent cells, which could be a promising strategy to attenuate vascular aging in cancer survivors previously treated with DOX [2]. Following the characterization of DOX-induced senescence, we determined the effects of several reported senolytics, including quercetin, dasatinib, fisetin, and ABT-263, across both cell lines. These senolytics disrupt different pro-survival pathways that are overexpressed by senescent cells, thereby inducing their apoptosis. Interestingly, the pro-survival mechanisms have been shown to vary across different senescent cell types. Consequently, senolytics that target these pathways can have distinct effects in different cell lines [16,26]. Dasatinib, a small-molecule inhibitor of tyrosine kinase, and quercetin, an inhibitor of the PI3K/AKT signaling pathway, were the earliest drugs that showed senolytic activity in vitro [15]. Interestingly, dasatinib was more effective in eliminating senescent preadipocytes than HUVECs [15,16,17], while quercetin exerted senolytic effects in senescent HUVECs but not in preadipocytes. Fisetin, a quercetin-related flavonoid, has also been shown to have senolytic activity in HUVECs but not preadipocytes by inhibiting the PI3K/AKT pathway [16]. ABT-263, a non-selective inhibitor of the BCL-2 family, targets the overexpressed anti-apoptotic pathways that allow senescent cells to resist apoptosis and maintain their viability [44]. Similarly, ABT-263′s senolytic activity was shown to be dependent on the cell type [17]. In the current study, dasatinib, quercetin, and fisetin demonstrated a lack of selective senolytic activity against DOX-induced senescent EA.hy926 cells and HUVECs. Since the senolytic effects has been shown to be dependent on the mechanism of senescence induction [45], the lack of senolytic activity of these drugs may be due to the DOX-induced senescence model that we used in our study compared to the irradiation-induced senescence in previous studies [15,17]. Only ABT-263 selectively induced the apoptosis of DOX-induced senescent HUVECs, but not EA.hy926 cells. The demonstrated effectiveness of different senolytics was in agreement with a recent study evaluating the sensitivity of radiation-induced senescent glioblastoma cells to different senolytics [46]. Similar to our findings in HUVECs, only the inhibition of BCL-xL, but not BCL-2 nor other senolytic targets, was able to demonstrate senolytic activity.

To elucidate the underlying mechanisms of the differential effects of ABT-263 in EA.hy926 cells and HUVECs, we measured the expression of the BCL-2 family, the main targets of ABT-263, across both cell lines. We demonstrated the differential expression of BCL-2 anti-apoptotic and apoptotic markers in EA.hy926 cells and HUVECs. While these markers were upregulated in DOX-induced senescent vs. control non-senescent HUVECs, no alterations were observed in senescent vs. non-senescent EA.hy926 cells following DOX treatment. The upregulation of BCL-2 family proteins is suggested to mediate the observed senolytic effects of ABT-263 in HUVECs.

Of note, BCL-2 protein expression was not detected in either control or DOX-treated EA.hy926 cells. Interestingly, previous studies have demonstrated a lack of expression of BCL-2 in A549 human lung carcinoma cells [47,48,49], the parent cells of EA.hy926. This suggests that EA.hy926 endothelial cells may have retained some characteristics of the A549 cancer cells, meaning they do not express BCL-2. This was confirmed by the diminished response of EA.hy926 cells to the selective BCL-2 inhibitor venetoclax compared to HUVECs. These findings suggest that BCL-2 is dispensable for the apoptotic effect of ABT-263 and its activity is primarily through the inhibition of BCL-xL. Interestingly, ABT-263 was previously reported to exert its senolytic activity via the disruption of the BAX/BCL-xL complex in tumor cells [50]. However, this mechanism is not fully supported by our current experimental data, and future research is warranted to delineate the exact mechanism of the senolytic effect of ABT-263 in HUVECs. Additionally, these findings align with previous RNA interference reports demonstrating that BCL-xL but not BCL-2 is necessary for the survival of senescent HUVECs [15] and senescent DOX-treated cancer cells [50].

## 5. Conclusions

In conclusion, our present study demonstrates that low concentrations of DOX induced relatively similar senescence phenotypes in immortalized EA.hy926 cells and primary HUVECs. However, the two cell lines responded differently to anti-BCL-2 family drugs such as ABT-263 and venetoclax. Hence, the effects of senolytics that target the anti-apoptotic pathways should be interpreted with caution in EA.hy926 cells. In addition to HUVECs, other models of primary ECs have been used to study endothelial senescence, including human microvasculature endothelial cells (HMVECs) [51], human aortic endothelial cells (HAECs) [12,52,53], and human coronary artery endothelial cells (HCAECs) [54]. Therefore, future studies are needed to determine whether our results in HUVECs are generalizable to other primary ECs.

## Figures and Tables

**Figure 1 cells-11-01992-f001:**
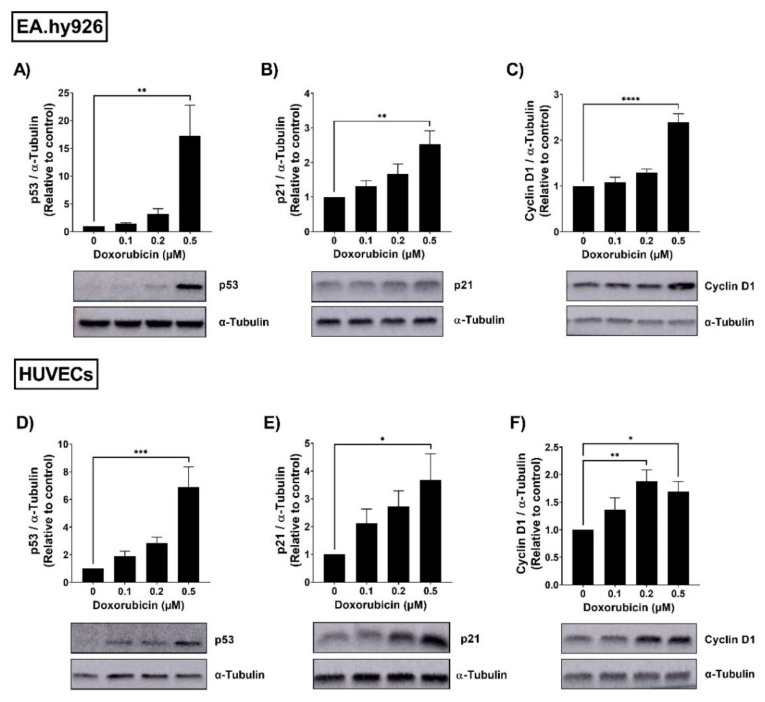
Doxorubicin induces the expression of senescence markers in a concentration-dependent manner in EA.hy926 cells and HUVECs. EA.hy926 human-endothelial-derived cells and HUVECs were treated with increasing concentrations of DOX (0.1 µM, 0.2 µM, and 0.5 µM) for 24 h. Thereafter, DOX was removed and cells were incubated in DOX-free media for 72 h. Then, cells were harvested and the total protein was extracted. Expression levels of senescence markers including p53, p21, and cyclin D1 in EA.hy926 cells ((**A**–**C**), respectively) and HUVECs ((**D**–**F**), respectively) were measured via Western blotting (*n* = 4–6). Representative images of Western blots are shown. Values were normalized to α-tubulin and expressed relative to control cells. Values are presented as means ± SEM. Data were analyzed by one-way ANOVA followed by Dunnet’s multiple comparisons test. Note: * *p* < 0.05, ** *p* < 0.01, *** *p* < 0.001, **** *p* < 0.0001.

**Figure 2 cells-11-01992-f002:**
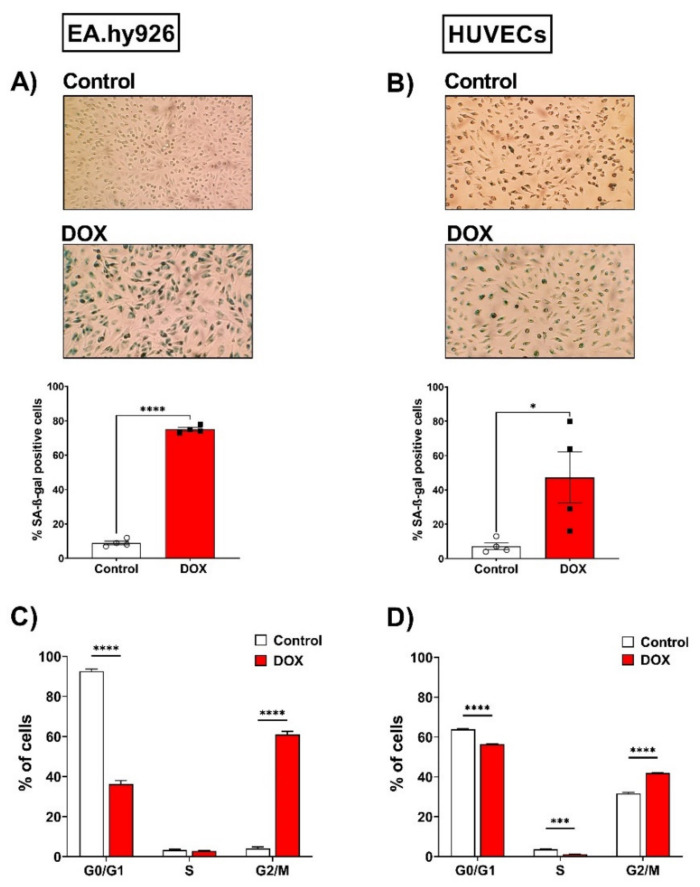
Doxorubicin triggers senescence in EA.hy926 cells and HUVECs, as demonstrated by the increased SA-β-gal activity and cell cycle arrest. EA.hy926 human-endothelial-derived cells and HUVECs were treated with DOX (0.5 µM) for 24 h. Thereafter, DOX was removed and cells were incubated with DOX-free media for 120 h then stained for SA-β-gal. Images of SA-β-gal staining of EA.hy926 cells (**A**) and HUVECs (**B**) are shown. The percentage of SA-β-gal positive cells were calculated. Data were analyzed via unpaired two-tailed t-test. In another set of experiments, cells were incubated with DOX-free media for 72 h and the cell cycle was analyzed by measuring the DNA content using the FACSCanto system. Percentages of each cell cycle phase in control and DOX-treated cells (*n* = 5–6) are shown for EA.hy926 (**C**) and HUVECs (**D**). Data are presented as means ± SEM. Data were analyzed via two-way ANOVA followed by Sidak’s post hoc test. Note: * *p* < 0.05, *** *p* < 0.001, **** *p* < 0.0001.

**Figure 3 cells-11-01992-f003:**
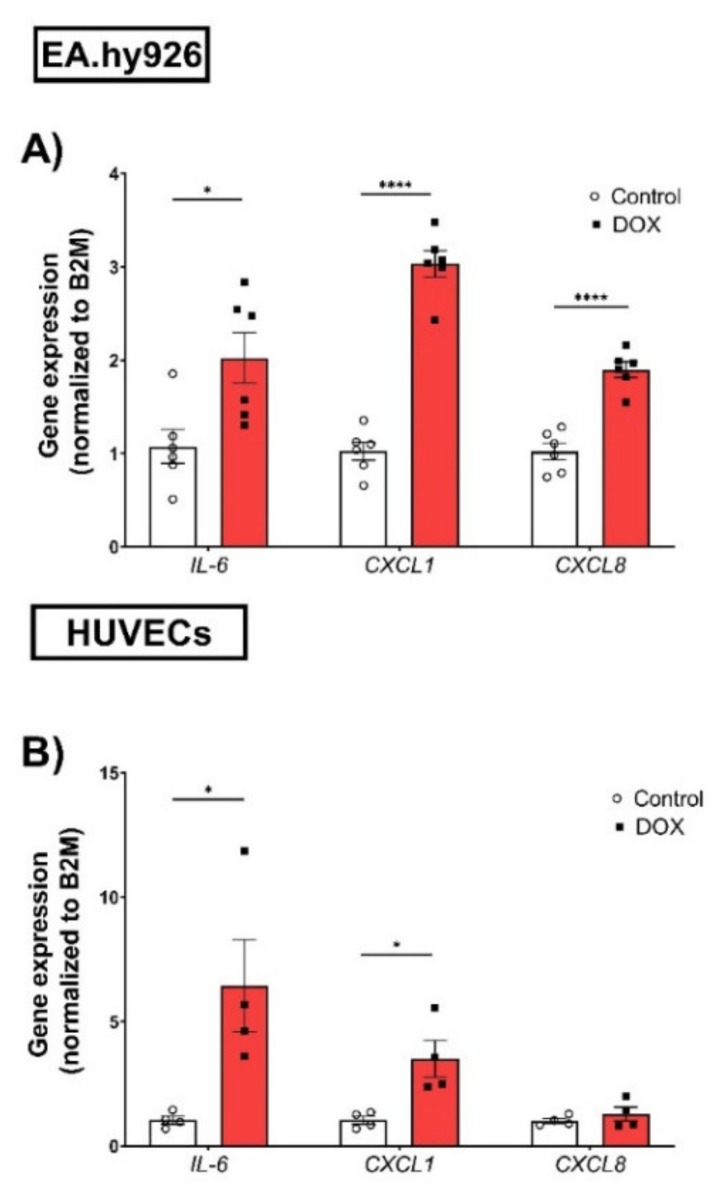
DOX induces the gene expression of SASP factors in both EA.hy926 cells and HUVECs. EA.hy926 human-endothelial-derived cells and HUVECs were treated with DOX (0.5 µM) for 24 h. Thereafter, DOX was removed and cells were incubated with DOX-free media for 72 h. The total RNA was then extracted and the mRNA expression of SASP factors including IL-6, CXCL1, and CXCL8 in (**A**) EA.hy926 cells (*n* = 6) and (**B**) HUVECs (*n* = 4) was determined by real-time PCR. Values were normalized to B2M and expressed relative to control cells. Values are shown as means ± SEM. Data were analyzed by unpaired two-tailed t-test. Note: **p* < 0.05, **** *p* < 0.0001.

**Figure 4 cells-11-01992-f004:**
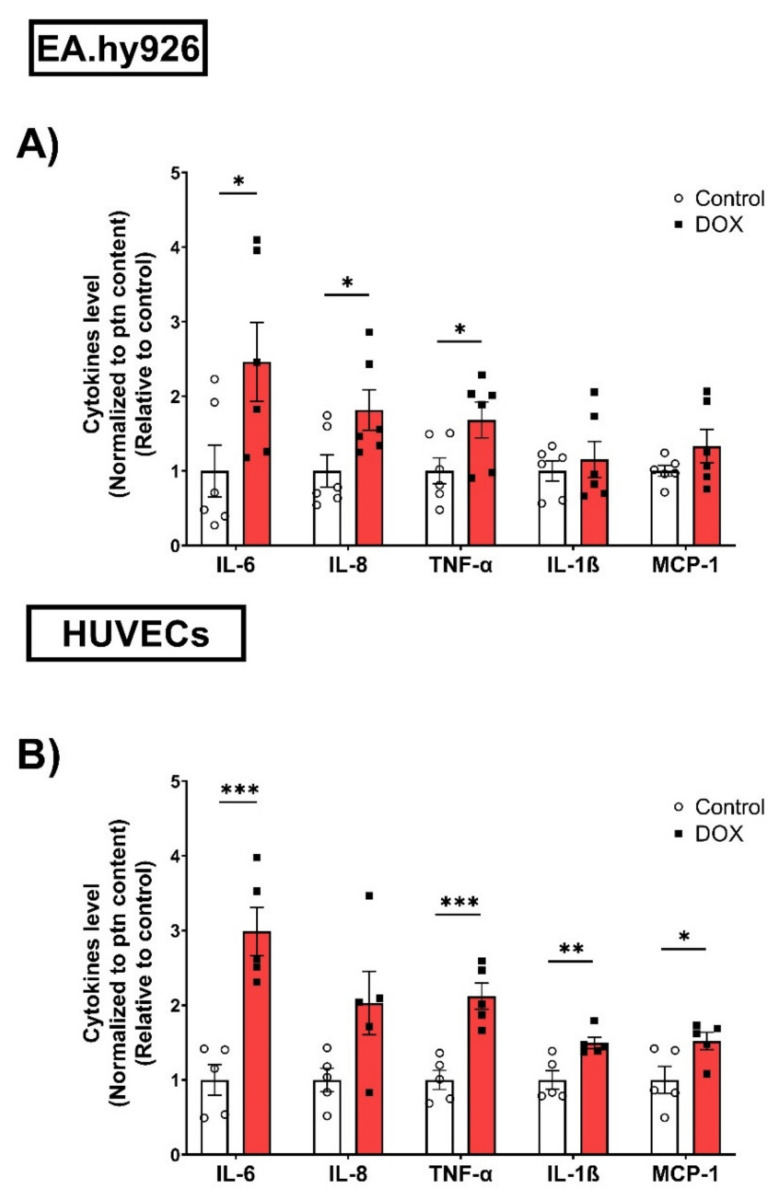
DOX induces the protein expression of SASP factors in conditioned media of EA.hy926 cells and HUVECs. EA.hy926 human-endothelial-derived cells and HUVECs were treated with DOX (0.5 µM) for 24 h. Thereafter, DOX was removed and cells were incubated with DOX-free media for 72 h. Conditioned media were collected and the expression of SASP factors including IL-6, TNF-α, IL-8, IL-1B, and MCP-1 in (**A**) EA.hy926 cells (*n* = 6) and (**B**) HUVECs (*n* = 5) was determined by Luminex. Values were normalized to the protein (ptn) content of the cells determined by BCA and the results are expressed as the fold expression relative to control cells. Values are shown as means ± SEM. Data were analyzed by unpaired two-tailed t-test. Note: * *p* < 0.05, ** *p* < 0.01, *** *p* < 0.001.

**Figure 5 cells-11-01992-f005:**
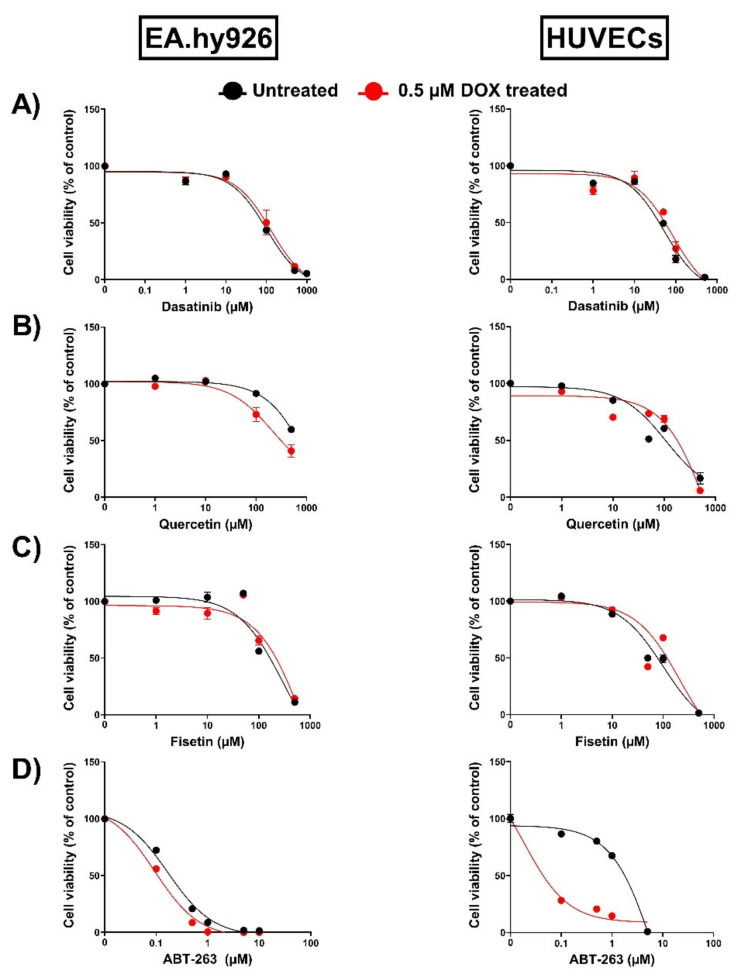
Assessment of the activity of multiple senolytics in EA.hy926 cells and HUVECs. Both EA.hy926 human-endothelial-derived cells and HUVECs were treated with DOX (0.5 µM) for 24 h to establish the senescence phenotype or left untreated. Three days post DOX exposure, DOX-treated and untreated cells were incubated with increasing concentrations of different senolytics including (**A**) dasatinib, (**B**) quercetin, (**C**) fisetin, and (**D**) ABT-263 for 24 h. Thereafter, the cell viability was measured using MTT assays in both cell lines. The cell viability was calculated relative to control wells and expressed as a percentage. Values are presented as means ± SEM.

**Figure 6 cells-11-01992-f006:**
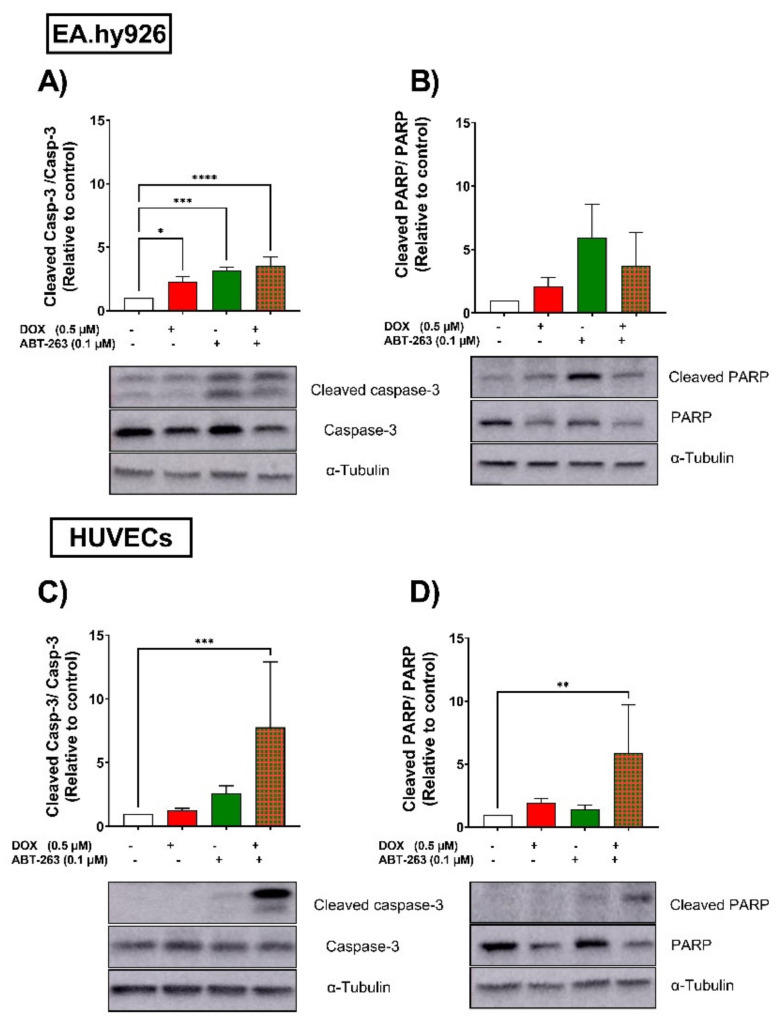
EA.hy926 and HUVEC cell lines respond differently to the senolytic drug ABT-263. Both EA.hy926 cells and HUVECs were treated with DOX (0.5 µM) for 24 h or left untreated (control cells). Three days post DOX exposure, DOX-treated and control cells were treated with ABT-263 (0.1 µM) for 6 h. Thereafter, expression levels of the apoptotic markers cleaved caspase-3 and cleaved PARP in EA.hy926 cells (**A**,**B**, respectively) and HUVECs (**C**,**D**, respectively) were measured via Western blotting (*n* = 4–6). Values are presented as means ± SEM. Data were analyzed by one-way ANOVA followed by Dunnet’s multiple comparisons test. Note: * *p* < 0.05, ** *p* < 0.01, *** *p* < 0.001, **** *p* < 0.0001.

**Figure 7 cells-11-01992-f007:**
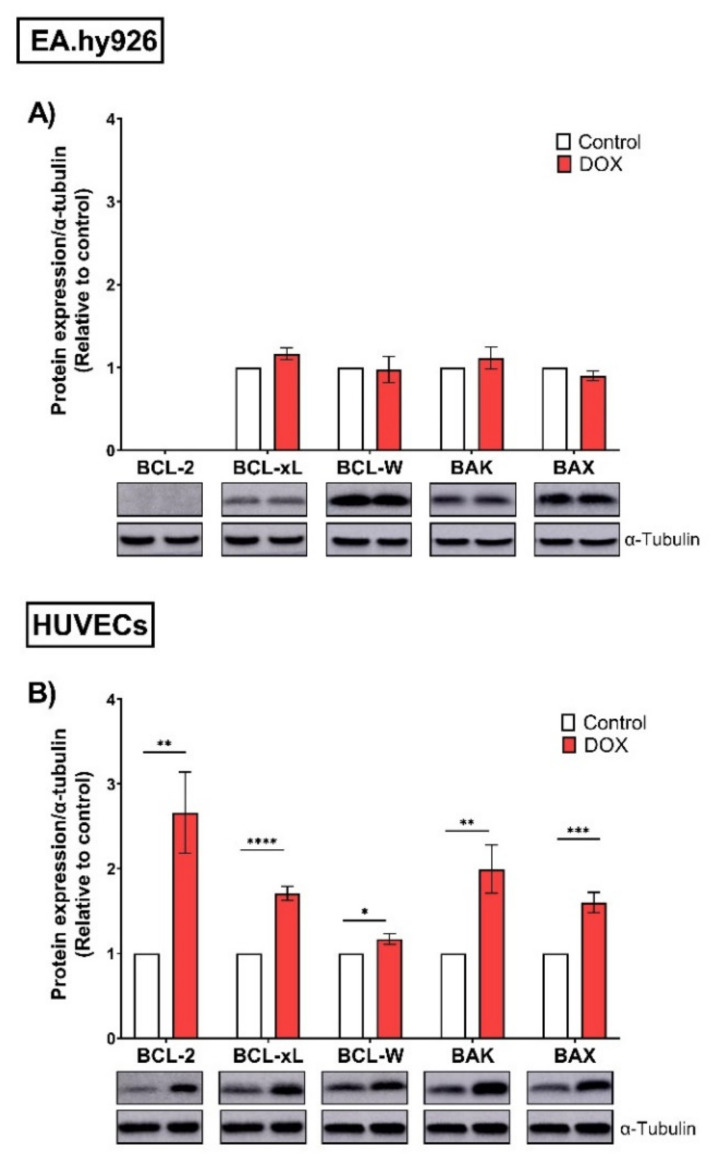
EA.hy926 and HUVECs demonstrate differential protein expression of the BCL-2 family following DOX treatment. Both EA.hy926 cells and HUVECs were treated with 0.5 µM DOX for 24 h. Three days post DOX exposure, cells were harvested and total protein was extracted. Thereafter, protein expression levels of the BCL-2 family including anti-apoptotic members (BCL-2, BCL-xL, and BCL-W) and pro-apoptotic members (BAK and BAX) in (**A**) EA.hy926 cells and (**B**) HUVECs were measured via Western blotting (*n* = 4–6). Representative images of Western blots are shown. Values were normalized to α-tubulin and expressed relative to control cells. Values are presented as means ± SEM. Data were analyzed by unpaired two-tailed t-test. Note: * *p* < 0.05, ** *p* < 0.01, *** *p* < 0.001, **** *p* < 0.0001.

**Figure 8 cells-11-01992-f008:**
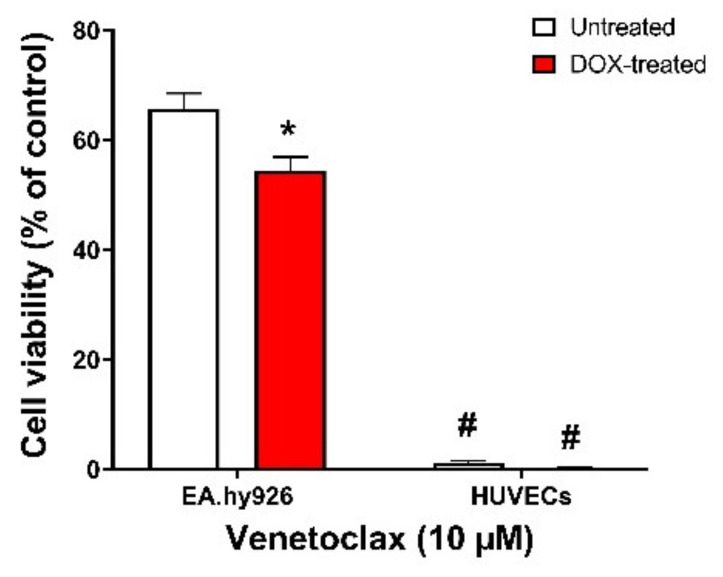
EA.hy926 cells and HUVECs respond differently to the BCL-2-selective inhibitor venetoclax. Both EA.hy926 cells and HUVECs were treated with DOX (0.5 µM) for 24 h to establish the senescence phenotype or left untreated. Three days post DOX exposure, DOX-treated and untreated cells were incubated with 10 µM venetoclax for 24 h. Thereafter, the cell viability was measured using MTT assays in both cell lines. The cell viability was calculated relative to control wells and expressed as a percentage. Note: * *p* < 0.05, compared to untreated treatment of the same cell line; # *p* < 0.05, compared to EA.hy926 cells of same treatment by two-way ANOVA with Tukey’s post hoc analysis. Values are presented as means ± SEM.

**Table 1 cells-11-01992-t001:** Primer sequences used in this study.

Gene	Forward Primer (5′–3′)	Reverse Primer (3′–5′)	Ref
*IL-6*	CCGGGAACGAAAGAGAAGCT	GCGCTTGTGGAGAAGGAGTT	[21]
*CXCL1*	GAAAGCTTGCCTCAATCCTG	CACCAGTGAGCTTCCTCCTC	[21]
*CXCL8*	CTTTCCACCCCAAATTTATCAAAG	CAGCAGAGCTCTCTTCCATCAGA	[21]
*B2M*	CACCCCCACTGAAAAAGATGAG	CCTCCATGATGCTGCTTACATG	[22]

## Data Availability

Not applicable.

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
