# Peer review of "EA.hy926 Cells and HUVECs Share Similar Senescence Phenotypes but Respond Differently to the Senolytic Drug ABT-263"

_cells, 2022, doi:10.3390/cells11131992_

Round 1

Reviewer 1 Report

Abdelgawad et al characterized doxorubicin-induced senescence phenotype in primary endothelial (HUVEC) and in immortalized endothelial-derived cells (EA.hy926), and demonstrated the similarities and differences in response of these cell types to different senolytic drugs. The authors evaluated the effect of four senolytic drugs in both endothelial cells and observed differences to ABT-263, due to differential expression of the BCL-2 family proteins. It is a well-designed and detailed study and can be accepted after the following two minor corrections:

1.     Edit the line 53 to include “with” by saying “cells by interfering with”

2.     Expand SASP in line 287, if that’s first occurrence of the word in the manuscript

Author Response

Reviewer 1:

Abdelgawad et al characterized doxorubicin-induced senescence phenotype in primary endothelial (HUVEC) and in immortalized endothelial-derived cells (EA.hy926), and demonstrated the similarities and differences in response of these cell types to different senolytic drugs. The authors evaluated the effect of four senolytic drugs in both endothelial cells and observed differences to ABT-263, due to differential expression of the BCL-2 family proteins. It is a well-designed and detailed study and can be accepted after the following two minor corrections:

Response: We would like to thank the respected reviewer for their comment.

  1. Edit the line 53 to include “with” by saying “cells by interfering with”

Response: edited in the revised manuscript.

  1. Expand SASP in line 287, if that’s first occurrence of the word in the manuscript

Response: expanded in the revised manuscript.

Reviewer 2 Report

This is a relatively straightforward study demonstrating differences in sensitivity to senolytics in two often used endothelial cell models responding to doxorubicin induced senescence. The authors utilize appropriate and clinically relevant concentrations of doxorubicin in their experiments. They demonstrate essentially no response to dasatinib, quercetin or fisetin, agents that have senolytic activity in aging related pathologies. They further demonstrate selective action of navitoclax against senescent HUVECs, but what might be considered off-target toxicity in the hybrid EA endothelial cell model. This is argued as being related to the induction of specific members of the Bcl-2 family in the HUVECs. While these observations are not entirely novel, these comparative studies should nevertheless prove informative for other laboratories interested in senescence and senolytics as potential therapeutic modalities. 

The assessment of proteins associated with the senescence associated secretory phenotype in conditioned media is considered a relevant experiment, since most laboratories simply assess expression of these factors in the cell.

Suggested modifications to the manuscript.

The studies relating to MAPK signaling pathways do not contribute much in the way of mechanistic insights and could probably be removed.

It is insufficient to utilize PARP and caspase 3 cleavage as indicators of apoptosis. The actual extent of apoptosis should be measured by standard assays such as PI/Annexin coupled with flow cytometry.

The conclusions relating to the pro-apoptotic and anti-apoptotic proteins are largely circumstantial in the absence of additional mechanistic experiments to assess their association under doxorubicin induced senescence and dissociation by navitoclax (and likely the lack of effects on these associations/dissociations by the fisetin, dasatinib and quercetin). These experiments, in addition to the more rigorous evaluation of apoptosis indicated above, would markedly strengthen this work.

The authors might want to emphasize the larger picture in terms of the conditions under which drugs such as fisetin and dasatinib/quercetin are and are not effective as senolytics. In this regard, a recent paper by Rahman M et al ( PMID 35191501) that assessed different senolytics in glioblastoma senescence models could be cited to bolster the argument relating to senescence inducers and senolytic sensitivity and selectivity.

Author Response

Reviewer 2:

This is a relatively straightforward study demonstrating differences in sensitivity to senolytics in two often used endothelial cell models responding to doxorubicin induced senescence. The authors utilize appropriate and clinically relevant concentrations of doxorubicin in their experiments. They demonstrate essentially no response to dasatinib, quercetin or fisetin, agents that have senolytic activity in aging related pathologies. They further demonstrate selective action of navitoclax against senescent HUVECs, but what might be considered off-target toxicity in the hybrid EA endothelial cell model. This is argued as being related to the induction of specific members of the Bcl-2 family in the HUVECs. While these observations are not entirely novel, these comparative studies should nevertheless prove informative for other laboratories interested in senescence and senolytics as potential therapeutic modalities.

The assessment of proteins associated with the senescence associated secretory phenotype in conditioned media is considered a relevant experiment, since most laboratories simply assess expression of these factors in the cell.

Response: We would like to thank the respected reviewer for their comment.

Suggested modifications to the manuscript.

The studies relating to MAPK signaling pathways do not contribute much in the way of mechanistic insights and could probably be removed.

Response: We would like to thank the reviewer for this suggestion. Although we think that these results offer important mechanistic insights into doxorubicin-induced senescence, we agree with the reviewer that they may not be very relevant to the current manuscript. Therefore, in order to keep the current manuscript concise and straightforward, the studies relating to MAPK signaling pathways have been removed from all sections.

It is insufficient to utilize PARP and caspase 3 cleavage as indicators of apoptosis. The actual extent of apoptosis should be measured by standard assays such as PI/Annexin coupled with flow cytometry.

Response: We would like to thank the reviewer for this suggestion. We agree that assessment of apoptosis using standard assay such as PI/Annexin may strengthen our conclusions. Unfortunately, although measuring PI/Annexin by flow cytometry is a straightforward technique, it is not established/optimized in our lab yet. We will work on establishing this technique for future studies. That being said, findings from previous studies of senolytics have shown complete correlation between the results of PI/ Annexin and the apoptotic markers cleaved-PARP (He, Li et al. 2020, Saleh, Carpenter et al. 2020) and cleaved-caspase 3 (Saleh, Carpenter et al. 2020). Therefore, we believe that measuring cleaved caspase-3 and cleaved PARP is strong evidence of apoptosis. Importantly, our apoptosis results match the viability assays, further strengthening our conclusions. In support of this argument, a number of leading studies in the field of senolytics depended on the caspase-3/7 activity as indication of apoptosis, without measuring PI/ Annexin by flowcytometry (Zhu, Tchkonia et al. 2016, Zhu, Doornebal et al. 2017).

The conclusions relating to the pro-apoptotic and anti-apoptotic proteins are largely circumstantial in the absence of additional mechanistic experiments to assess their association under doxorubicin induced senescence and dissociation by navitoclax (and likely the lack of effects on these associations/dissociations by the fisetin, dasatinib and quercetin). These experiments, in addition to the more rigorous evaluation of apoptosis indicated above, would markedly strengthen this work.

Response: We would like to thank the reviewer for this suggestion. Unfortunately, we could not perform this experiment due to the low amount of protein yield that was totally consumed by our extensive western blotting studies. Since ABT-263 decreased cell viability, the obtained protein concentrations were just enough to run western blotting for the apoptotic markers cleaved caspase-3 and cleaved PARP.

Additionally, in a landmark study, ABT-263 has already been shown to exert its senolytic activity via the disruption of the BAX/BCL-xL complex (Saleh, Carpenter et al. 2020). Therefore, we think that this important mechanistic aspect of navitoclax has already been discovered. Since the rest of screened senolytics demonstrate lack of senolytic activity and considering that they target other pathways than BCL-2 family, we did not evaluate their effects on the association/dissociation of pro-apoptotic and anti-apoptotic proteins.

Please refer to lines 503-507:

This is in agreement with a previous RNA interference report demonstrating that BCL-xL but not BCL-2 is necessary for the survival of senescent HUVECs (Zhu, Tchkonia et al. 2015) and senescent DOX-treated cancer cells (Saleh, Carpenter et al. 2020). Mechanistically, ABT-263 has been shown to exert its senolytic activity via the disruption of the BAX/BCL-xL complex (Saleh, Carpenter et al. 2020)”.

The authors might want to emphasize the larger picture in terms of the conditions under which drugs such as fisetin and dasatinib/quercetin are and are not effective as senolytics. In this regard, a recent paper by Rahman M et al (PMID 35191501) that assessed different senolytics in glioblastoma senescence models could be cited to bolster the argument relating to senescence inducers and senolytic sensitivity and selectivity.

Response: We would like to thank the reviewer for this comment. We have cited the suggested article in the discussion section

Please, refer to lines 485-488:

The demonstrated effectiveness of different senolytics was in agreement with a recent study evaluating the sensitivity of radiation-induced senescent glioblastoma cells to different senolytics (Rahman, Olson et al. 2022). Similar to our findings in HUVECs, only the inhibition of BCL-xL but not BCL-2 nor other senolytic targets was able to demonstrate senolytic activity.” 

Reviewer 3 Report

Abdelgawad et al., have produced a mechanistically interesting paper that aims to investigate the potential routes by which certain senolytics function in response to doxorubicin induced senescence. The authors add to the growing literature indicating that Navitoclax is a potent and preferential target for senescent primary HUVEC cells and also incorporate the inclusion of an immortalised cell line to the story. However, I understand the desire and benefits of using immortalised cell lines in certain circumstances, but I fail here to comprehend why in this particular scenario this is the case. We all recognise the difficulties/drawbacks with working with primary cells, but given the area in question (i.e. the vasculature) their use/inclusion is imperative. By highlighting the difference in responses between HUVEC and EA.hy926 it solidifies to need to use other primary cells form within the vasculature. For example, Human Aortic Endothelial cells (HAECs) and Human Microvasculature Endothelial cells (HMVECs). I would recommend moving away from the use of immortalised cells lines in this area, especially given the fundamental difference in transcriptomics highlighted. Overall, the paper is well written and the experimental studies are well designed.

Author Response

Reviewer 3:

Abdelgawad et al., have produced a mechanistically interesting paper that aims to investigate the potential routes by which certain senolytics function in response to doxorubicin induced senescence. The authors add to the growing literature indicating that Navitoclax is a potent and preferential target for senescent primary HUVEC cells and also incorporate the inclusion of an immortalised cell line to the story. However, I understand the desire and benefits of using immortalised cell lines in certain circumstances, but I fail here to comprehend why in this particular scenario this is the case. We all recognise the difficulties/drawbacks with working with primary cells, but given the area in question (i.e. the vasculature) their use/inclusion is imperative. By highlighting the difference in responses between HUVEC and EA.hy926 it solidifies to need to use other primary cells form within the vasculature. For example, Human Aortic Endothelial cells (HAECs) and Human Microvasculature Endothelial cells (HMVECs). I would recommend moving away from the use of immortalised cells lines in this area, especially given the fundamental difference in transcriptomics highlighted. Overall, the paper is well written and the experimental studies are well designed.

Response: We would like to thank the respected reviewer for their comment. We totally agree with the reviewer that it is critical to use primary ECs to characterize vascular senescence, a notion that is further strengthened by our current findings. Since immortalized cells are still being used as models for endothelial senescence (Upreti, Koonce et al. 2010, Ghosh, Rai et al. 2016, Altieri, Murialdo et al. 2017, Sun, Dou et al. 2018, Sun, Ghosh et al. 2019), we think it is important to demonstrate their similarities/differences from primary endothelial cells.

In the revised manuscript, we removed the following sentence “This suggests that both HUVECs and EA.hy926 cells can be used to characterize DOX-induced senescence”, which may have implied that we advocate for the use of both cell lines.

We also agree with the respected reviewer that the next steps following this work should be comparing different primary endothelial cells in both their senescence phenotype and their response to different senolytics. We have updated the conclusion of our manuscript to add this as a future direction for our work.

Please, refer to lines 513-518:

In addition to HUVECs, different models of primary ECs have been used previously to study senescence including human microvasculature endothelial cells (HMVECs) (Heo, Kim et al. 2016), human aortic endothelial cells (HAECs) (Park, Kim et al. 2016, Casella, Munk et al. 2019, Chen, Holder et al. 2021), and human coronary artery endothelial cells (HCAECs) (Sun, Ghosh et al. 2019). Therefore, future studies are needed to determine if our results in HUVECs are generalizable to other primary ECs”.

Round 2

Reviewer 2 Report

Although the authors were unable to perform some of the experiments indicated in the previous review, the studies in this manuscript are nevertheless worthy of publication.

Author Response

We would like to thank the respected reviewer for their valuable comments.

Reviewer 3 Report

.

Author Response

(The authors gave the same response as above.)
